# Acrylamide Exposure and Cardiovascular Risk: A Systematic Review

**DOI:** 10.3390/nu16244279

**Published:** 2024-12-11

**Authors:** Diana María Mérida, Jimena Rey-García, Belén Moreno-Franco, Pilar Guallar-Castillón

**Affiliations:** 1Department of Preventive Medicine and Public Health, School of Medicine, Universidad Autónoma de Madrid, 28029 Madrid, Spain; diana.merida@estudiante.uam.es; 2Department of Pharmacoepidemiology and Biostatistics, Fundación Teófilo Hernando, 28290 Las Rozas de Madrid, Spain; 3CIBERESP (CIBER of Epidemiology and Public Health), 28029 Madrid, Spain; 4Department of Internal Medicine, Hospital Universitario Rey Juan Carlos, IIS-FJD, 28933 Móstoles, Spain; 5Instituto de Investigación Sanitaria Aragón, Hospital Universitario Miguel Servet, 50009 Zaragoza, Spain; mbmoreno@unizar.es; 6Department of Preventive Medicine and Public Health, Universidad de Zaragoza, 50009 Zaragoza, Spain; 7CIBERCV (CIBER of Cardiovascular Diseases), 28029 Madrid, Spain; 8IMDEA-Food Institute, CEI UAM+CSIC, Carretera de Cantoblanco 8, 28049 Madrid, Spain

**Keywords:** acrylamide, cardiovascular, morality, diabetes, lipids, obesity

## Abstract

*Background/Objectives:* Acrylamide is a food contaminant formed during high-temperature cooking processes, leading to unintentional human exposure. Diet is the primary source for non-smokers, with potatoes, cereals, and coffee being the main contributors. While animal studies have demonstrated that acrylamide is neurotoxic, genotoxic, mutagenic, and cardiotoxic, its effects on human cardiovascular health remain poorly understood. This study aimed to evaluate the association between acrylamide exposure and cardiovascular risk. *Methods:* A comprehensive literature search was conducted across four databases without restrictions on publication year or language (last search: 1 July 2024). The risk of bias was assessed using the Joanna Briggs Institute critical appraisal tools. *Results:* In total, 28 studies were included, predominantly from the US NHANES sample and with cross-sectional designs. Higher acrylamide exposure was associated with an increased risk of cardiovascular mortality but was inversely associated with glucose and lipid levels, as well as key cardiovascular risk factors such as diabetes, obesity, and metabolic syndrome. Conversely, glycidamide—acrylamide’s most reactive metabolite—was positively associated with elevated glucose and lipid levels, higher systolic blood pressure, and increased obesity prevalence. *Conclusions:* These findings suggest that the adverse cardiovascular effects of acrylamide may be mediated by its conversion to glycidamide. Further research is necessary to fully elucidate the impact of acrylamide on cardiovascular health. Meanwhile, public health efforts should continue to focus on mitigation strategies within the food industry and raising public awareness about exposure.

## 1. Introduction

Acrylamide is a chemical substance used in gel electrophoresis, as well as in the production of crude oil, cosmetic additives, and other substances such as polyacrylamide, with applications in pulp and paper production, agriculture, food processing, mining, and as a flocculant in wastewater treatment. It is also present in tobacco smoke [1,2]. In addition, acrylamide has been classified as a food contaminant formed by the Maillard reaction between asparagine and sugars reducing during food processing at temperatures above 120 °C [3]. For this reason, diet is the main source of exposure in the non-smoking population, with potato products, cereal products, and coffee being the major contributors [3,4]. Therefore, humans are not only exposed to acrylamide through tobacco smoke, but also through other unintentional exposures via food, dermal contact, and inhalation [5].

After exposure, acrylamide reaches the systemic blood circulation and is metabolized in the liver via the mercapturic acid pathway and partially metabolized by cytochrome P450 2E1 (CYP2E1) to the more reactive epoxide glycidamide [6]. Glycidamide causes greater genotoxic and mutagenic damage than acrylamide [7]. The proportion of acrylamide converted to glycidamide varies according to activity, which is influenced by genetic factors, age, alcohol consumption, diet, and environmental exposure [6]. Both acrylamide and glycidamide are electrophilic and bind covalently to proteins such as hemoglobin (Hb) and DNA [8]. Hb adducts of acrylamide (HbAA) and of glycidamide (HbGA) provide a good estimate of long-term exposure. In contrast, the use of urinary mercapturic acid metabolites as biomarkers provides insight into short-term exposure to acrylamide (i.e., N-acetyl-S-(2-carbamoylethyl)-l-cysteine (AAMA) and N-acetyl-S-(2-carbamoyl-2-hydroxyethyl)-l-cysteine (GAMA)) [9]. The conversion of acrylamide to glycidamide is represented by the HbGA/HbAA ratio [10].

In animal studies, acrylamide has been shown to be neurotoxic, genotoxic, mutagenic, and cardiotoxic [11], and the carcinogenic activity of acrylamide depends on its metabolism to glycidamide [8]. The association between acrylamide exposure and cancer has been extensively studied in humans, but the results are inconclusive for most types of cancer [12], with the possible exception of gynecological cancers [13]. Research on the association between acrylamide exposure and cardiovascular disease (CVD) in humans is scarce, and current evidence is mainly based on cross-sectional studies with mixed results. Therefore, the aim of this review was to examine the association between acrylamide exposure and CVD mortality, CVD risk, and cardiovascular risk factors.

## 2. Materials and Methods

This systematic review was conducted according to the Preferred Reporting Items for Systematic Reviews and Meta-Analysis (PRISMA) guidelines [14]. The protocol is registered in the International Prospective Register of Systematic Reviews (PROSPERO) (registration number: CRD42023488410) [15].

### 2.1. Search Strategy

Four different databases were used for the literature search: Clarivate (Web of Science), Medline, PubMed, and Scopus, with no restrictions on the year of publication or language. The last search was updated on 1 July 2024. The search strategy included the following keywords: “acrylamide” or “glycidamide” and “humans” and (“cardiovascular” or “mortality” or “myocardial infarction” or “coronary heart disease” or “stroke” or “cerebrovascular” or “atherosclerosis” or “arterial injury” or “hypertension” or “diabetes” or “obesity” or “metabolic syndrome” or “insulin resistance” or “dyslipidemia” or “cholesterol” or “triglycerides”).

### 2.2. Study Selection

The study selection was performed by two of the co-authors (DMM and PG-C). After removing duplicates, the title and abstract were screened to identify eligible studies. After screening, full text reading was performed to select articles based on the inclusion criteria: (a) studies conducted in humans; (b) studies analyzing acrylamide exposure measured in urine (AAMA and GAMA) or blood (HbAA and HbGA) or from dietary intake; (c) studies providing association estimates between acrylamide exposure and cardiovascular risk factors, reporting OR, HR, RR, or linear regression coefficients (β); and (d) studies with any epidemiological design (longitudinal or cross-sectional). Studies were excluded if they were conducted in pregnant women and evaluated postnatal outcomes, did not report primary data, or evaluated other CVDs (e.g., peripheral arterial disease, rheumatic heart disease, etc.). Eligible studies were also identified by reviewing the reference lists of the selected studies. In the case of duplicate original data, the article with the most comprehensive information was included.

### 2.3. Outcomes

The main outcomes of this review were related to (a) CVD mortality and CVD risk; (b) diabetes mellitus (DM) and glucose metabolism; (c) dyslipidemia and lipid metabolism; (d) obesity and body composition; (e) hypertension (HTN) and blood pressure; and (f) metabolic syndrome (MetS).

### 2.4. Data Extraction

The following information was extracted from the studies: first author, year of publication, country and sample, sample size, age, percentage of men, acrylamide exposure (urine, blood, or dietary intake), outcome definition, and covariates in the fully adjusted model. The unit of comparison and the results of the association estimates (OR, HR, or RR) and linear regression coefficients (β) from the fully adjusted model were also extracted. Data extraction was performed by two of the co-authors (DMM and PG-C).

### 2.5. Risk of Bias

The risk of bias of the included studies was assessed using the Joanna Briggs Institute (JBI) critical appraisal tools [16]. The critical appraisal tool for cohort studies includes 11 questions with “Yes”, “No”, “Unclear” or “Not applicable” answers, and the tool for cross-sectional studies includes 8 questions. The overall risk of bias was established according to the percentage of “Yes” answers (the higher the score, the lower the risk of bias). The following cut-off points were used to determine the risk of bias of the studies: high risk of bias was considered if the score was less than 49%, moderate risk if the score was 50–69% and low risk if the score was more than 70%. Any discrepancies between the two co-authors (DMM and PG-C) were resolved by consensus.

### 2.6. Data Synthesis

The results were organized based on the primary outcome and for smokers and non-smokers, if this information was available. A meta-analysis could not be performed due to the heterogeneity of the studies. To better communicate the results, not only are the central estimators reported, but also the direction of the association using arrows (e.g., ↑ = higher prevalence, ↓ = lower prevalence).

## 3. Results

### 3.1. Search Results

The literature search identified 2530 articles, of which 984 were duplicates and 1546 were screened (Figure 1). After screening, 1485 articles were excluded based on the title and abstract, 61 were assessed for eligibility, and 11 were added from references. Of a total of 72 articles, 44 did not meet the inclusion criteria, and the reasons for exclusion are detailed in Appendix A. Finally, 28 articles were included in this systematic review [17,18,19,20,21,22,23,24,25,26,27,28,29,30,31,32,33,34,35,36,37,38,39,40,41,42,43,44].

### 3.2. Characteristics of the Included Studies

The articles were conducted in the USA (*n* = 22), China (*n* = 2), France (*n* = 1), Taiwan (*n* = 1), and Iran (*n* = 2), with 8 cohort studies and 20 cross-sectional studies. The sample sizes ranged from 696 to 72,585 participants. Most studies included participants aged ≥20 years. Two studies included adolescents [30,40], one study included participants with hyperglycemia [20], and two studies were conducted in men [17] and women [21] only. In most studies, acrylamide exposure was measured in blood (*n* = 12, corresponding to NHANES data) and urine (*n* = 12), while dietary exposure (*n* = 2) and cumulative occupational exposure (*n* = 2) were also considered (Table 1).

The mean HbAA ranged from 47.7 (US adults) to 65.0 pmol/g Hb (US adolescents) and HbGA ranged from 42.2 to 60.4 pmol/g Hb (US adults). Mean dietary acrylamide intake ranged from 32.6 (France) to 57 µg/day (Iran).

### 3.3. Risk of Bias

Most studies had a low (*n* = 23), moderate (*n* = 4), or high (*n* = 1) risk of bias (Appendix A). The exposure measurement was performed in a valid and reliable way in most of the studies (urine or blood), except when dietary exposure was analyzed. Finally, for some of the cohort studies, follow-up was not long enough to examine the incidence of specific outcomes.

### 3.4. Association Between Acrylamide Exposure and CVD Mortality and CVD Risk

For CVD mortality, causes of death were defined according to ICD 8–10th. Six studies were included for this outcome [17,18,20,21,22,23] (Table 2). Among US male workers potentially exposed to acrylamide, a 12% lower risk of CVD mortality than expected was observed over the period 1925–2002 [17]. Comparing extreme categories of exposure, among participants with hyperglycemia from the NHANES sample, HbAA was associated with an 84% higher risk of CVD mortality [20]. In French women, dietary acrylamide intake was also associated with a 33% higher risk of CVD mortality [21]. And in Iran, AAMA was associated with a 68% higher risk of ischemic heart disease (IHD) mortality [23].

For CVD risk, four studies were included (partially overlapping) [25,26,27,28]. In the NHANES sample, it was observed that AAMA was associated with a 54% higher prevalence of self-reported CVD [28]. A study from China showed that those participants who were in the highest quartile of exposure to AAMA and GAMA had a significantly higher 10-year CVD risk (calculated using an algorithm) of 47% and 67%, respectively, compared with those in the lowest quartile [25] (Table 2 and Figure 2).

### 3.5. Association Between Acrylamide Exposure and Diabetes and Glucose Metabolism

In most studies, glucose was analyzed as a continuous variable, and some also reported HbA1c, insulin, and HOMA-IR. DM was defined as fasting plasma glucose ≥ 126 mg/dL or 2 h plasma glucose ≥ 200 mg/dL, or use of medication for DM. Of the eight studies evaluating glucose metabolism [29,30,31,32,33,42,43,44], five analyzed data from different NHANES cycles (the samples partially overlapped) [29,32,42,43,44] (Table 3).

In China, those in the highest quartile of AAMA and ΣUAAM had higher glucose levels of about 0.20 and 0.17 mmol/L, respectively, compared with those in the lowest quartile [31]. In the NHANES sample, each unit increase in HbGA and HbGA/HbAA was associated with higher glucose levels of 2.51 and 3.12 mg/dL, respectively (Figure 3). HbAA and HbAA + HbGA were inversely associated with glucose. For HbA1c, HbAA was inversely associated, whereas HbGA and HbGA/HbAA were directly associated [43].

In the NHANES sample, HbAA was associated with a 29% lower prevalence of DM, and HbGA/HbAA with a 95% higher prevalence when comparing the extreme quartiles [32]. 

### 3.6. Association Between Acrylamide Exposure and Dyslipidemia and Lipid Metabolism

Lipid metabolism was assessed with total cholesterol (TC), HDL, triglycerides (TG) and LDL as continuous variables. For these outcomes, eight studies were included [29,30,34,35,40,42,43,44], six of which analyzed data from the NHANES (partially overlapping) [29,34,35,40,42,43,44] (Table 4).

In US adolescents, HbGA was associated with higher TC (2.83 mg/dL), but not associated with other serum lipids [40] (Figure 3). In US adults, HbGA was directly associated with TC (6.60 mg/dL), LDL (7.47 mg/dL), and TG (16.86 mg/dL), but inversely associated with HDL (−4.20 mg/dL). The results for HbAA were completely inverse to those for HbGA. The results for HbGA/HbAA were similar to those for HbGA [43]. Also, AAMA was associated with higher HDL levels (1.28 mg/dL) [35].

### 3.7. Association Between Acrylamide Exposure and Obesity and Body Composition

In most of the studies, obesity was considered as dichotomous. General obesity (GO) was defined as body mass index (BMI) ≥ 30 kg/m^2^ and abdominal obesity (AO) as waist circumference (WC) >102 cm in men and >88 cm in women. A total of nine studies were included [29,30,36,37,38,39,42,43,44], eight of which analyzed data from the NHANES (partially overlapping) [29,36,37,38,39,42,43,44] (Table 5).

In US adults, HbAA was associated with a 64% lower prevalence of AO, whereas HbGA and HbGa/HbAA were associated with a 78% and 95% higher prevalence of AO [43] (Figure 2). AAMA was associated with a 21% lower prevalence of AO and GO [39].

### 3.8. Association Between Acrylamide Exposure and Hypertension and Blood Pressure

A total of five studies were included for this outcome [40,41,42,43,44], two of which partially overlapped [42,43] (Table 6). HTN was defined as blood pressure ≥ 130/80 mmHg or antihypertensive treatment. In the NHANES sample, HbAA and HbGA were not significantly associated with HTN in either adolescents or adults [40,43], but AAMA was associated with a 40% lower HTN prevalence [44]. When systolic blood pressure (SBP) was considered as a continuous variable, HbGA was associated with an increase of about 0.49 mmHg in US adolescents [40] and HbAA + HbGA with an increase of 1.29 mmHg in adults [43] (Figure 3).

### 3.9. Association Between Acrylamide Exposure and Metabolic Syndrome

MetS was defined by the NCEP-ATP III (three or more criteria) (Table 7). Three studies were included for this outcome [42,43]. In the NHANES sample, HbAA and AAMA were associated with a 40% and 22% lower prevalence of MetS, respectively [43,44], whereas HbGA/HbAA was associated with a 61% higher prevalence of MetS when comparing the extreme quartiles [43].

### 3.10. Association Between Acrylamide Exposure and Cardiovascular Risk Factors in Non-Smokers

A total of nine studies reported results in non-smokers [19,23,24,25,32,33,37,41,43] (Appendix A). When comparing the extreme quartiles, HbGA and HbGA/HbAA were associated with a 78% and 88% lower risk of CVD mortality, respectively [19]. In contrast, AAMA was associated with a 2-fold higher risk of IHD mortality [23], and when assessing 10-year CVD risk, AAMA, GAMA, and ΣUAAM were associated with 50%, 73%, and 55% higher risk, respectively [25].

For metabolic diseases, comparing the extreme quartiles of exposure, HbAA was associated with a 54% lower prevalence of DM [32]. HbGA/HbAA was associated with a 55% higher prevalence of DM [32], a 3-fold higher prevalence of GO [37], and a 76% higher prevalence of MetS [43].

### 3.11. Association Between Acrylamide Exposure and Cardiovascular Risk Factors in Smokers

A total of seven studies reported results for smokers [23,24,32,33,37,41,43] (Appendix A). Among those in the highest quartile of exposure, HbAA was associated with a 3.7-fold increased risk of self-reported CVD prevalence [24] and a 33% lower prevalence of DM [32] compared with those in the lowest quartile. HbGA/HbAA was associated with a 59% lower prevalence of CVD [24], a 2-fold higher prevalence of DM [32], and a 2.5-fold higher prevalence of GO [37].

## 4. Discussion

This systematic review included 28 articles, 19 of which were conducted on NHANES samples, some with partially overlapping populations. It was found that higher acrylamide exposure was associated with a higher risk of CVD mortality and CVD risk, but inversely associated with most cardiovascular risk factors (lower levels of glucose, HbA1c, TC, TG, and LDL, and lower prevalence of DM, obesity, HTN, and MetS), as well as positively associated with HDL levels. However, higher exposure to glycidamide (its most reactive metabolite) was positively associated with 10 y CVD risk and most cardiovascular risk factors (higher levels of glucose, HbA1c, TC, TG, LDL, SBP, and higher prevalence of AO), and inversely associated with HDL levels. Few studies performed stratified analyses based on smoking status.

Another interesting finding was that US adolescents had the highest HbAA levels (mean 65.0 pmol/g Hb) compared to US adults, although adolescents did not have the highest HbGA concentrations (low conversion from HbAA to HbGA). It is important to highlight that the percentage of conversion from acrylamide to glycidamide is influenced by the expression of liver CYP2E1, which is induced by alcohol consumption, drugs, physical development, hormonal status [6], obesity, and diabetes [45]. This could explain the lower conversion rate to glycidamide observed in US adolescents. It has also been shown that the consumption of fried potatoes, doughnuts, hot dogs, popcorn, and nachos is responsible for higher acrylamide exposure in adolescents [46,47,48], along with sociodemographic factors [47] associated with smoking and the consumption of ultra-processed foods. The increase in the consumption of ultra-processed foods in recent decades, especially among adolescents, remains a public health concern [49,50].

Several mechanisms may explain the association between acrylamide exposure and CVD risk. For instance, acrylamide may act as an obesogenic agent by inducing fat accumulation and increasing the expression of PPARγ in male mice [51,52]. The nuclear receptor PPARγ is expressed primarily in adipose tissue, hematopoietic cells, and colon cells [53]. PPARγ regulates adipogenesis, lipid metabolism, inflammation, and insulin sensitivity [54]. In female rats, acrylamide exposure significantly increased glucose levels, impaired glucose tolerance, and decreased hepatic glycogen content. Acrylamide has been shown to promote gluconeogenesis and glycogenolysis and to decrease glycolysis by up- or down-regulating specific genes involved in glucose metabolism [55].

Some outcomes show contradictory results depending on the biomarker used to assess acrylamide exposure. For instance, AAMA was associated with increased glucose levels, whereas HbAA was associated with decreased glucose levels. This discrepancy may arise from the fact that HbAA reflects long-term exposure to acrylamide, whereas AAMA reflects short-term exposure [9]. Furthermore, markers of acrylamide and its metabolite glycidamide often show opposite associations for most outcomes. In our findings, acrylamide was negatively associated with most cardiovascular risk factors, whereas glycidamide was positively associated with most of them. This highlights the critical role of the conversion of acrylamide to glycidamide, as glycidamide is more toxic and induces greater oxidative damage, which may explain its adverse cardiovascular associations.

The social relevance of studying acrylamide exposure is that it is a ubiquitous food processing contaminant to which the entire population is unintentionally exposed throughout life, including during the prenatal period, childhood, adolescent development, and adulthood. Furthermore, exposure to acrylamide comes not only from ultra-processed foods, but also from home-cooked and restaurant foods [56], even when using new cooking methods such as air frying [57].

Given the above mentioned, finding ways to mitigate acrylamide production and consumption is one of the most important areas of research in the food industry. The Commission Regulation (EU) 2017/2158 of 20 November 2017 published mandatory acrylamide mitigation measures for the catering and restaurants industry [58], and in the US, FDA guidelines have been issued for this purpose [59]. However, the population is largely unaware of acrylamide production in home-cooked foods and its presence in their daily diet. Moreover, browned foods are mistakenly perceived as more appealing and flavorful. For this reason, the Spanish Agency for Consumer Affairs, Food Safety and Nutrition (AECOSAN) developed recommendations for consumers to minimize the presence of acrylamide in home-cooked foods (e.g., reducing the final temperature and cooking time of foods, mainly for potatoes, biscuits, bread, and breaded foods) [60]. The food industry also developed the “Acrylamide Toolbox 2019” to control the presence of acrylamide in different food production processes [61].

This review has several strengths. First, it is the first study to synthesize the evidence on the association between acrylamide exposure and multiple cardiovascular outcomes. Second, it clearly describes the evidence for this association to date, without omitting negative findings. Third, the results encourage other researchers to provide evidence in this area to clarify the effects of acrylamide on cardiovascular health. Finally, these results are also of great interest for public health, highlighting the control of environmental and food contaminants as a strategy for CVD prevention. This is not only a matter for public health policy, but also for the food industry. As home-cooked food is also one of the most common sources of exposure, educating the population about healthier eating habits and cooking methods is an important part of these public health policies. The main conclusions of this review are summarized in Table 8.

There are also some limitations. First, most of the studies included in this review are mainly based on the US NHANES, which may limit the generalizability and applicability of the findings to other populations. Second, although interest in acrylamide exposure has increased in recent years, the current evidence on its associations with CVD is scarce, fragmented, not standardized, and yields mixed results. Also, most of the results were not stratified according to tobacco consumption, which could be an important confounding factor. In addition, it seems that acrylamide and glycidamide are not associated in the same direction with certain cardiovascular risk factors. Thirdly, a meta-analysis could not be performed because of the differences in outcome measures and statistical approaches, the partly overlapping populations, and the lack of studies. Finally, except for mortality, most of the conclusions are based on a small number of studies and cross-sectional designs that do not allow us to establish causality.

## 5. Conclusions

Acrylamide exposure is associated with a higher risk of CVD mortality, 10-year CVD risk, and CVD prevalence. However, it is inversely associated with the prevalence of DM, GO, AO, HTN, and MetS. In addition, acrylamide is negatively associated with glucose, HbA1c, and lipid levels (TC, TG, and LDL) and positively associated with HDL. In contrast, glycidamide—its most reactive metabolite—is associated with adverse outcomes, including higher 10-year CVD risk, higher prevalence of AO, increased glucose and HbA1c levels, increased lipid levels (TC, TG, and LDL), higher SBP, and lower HDL levels. These findings suggest that the cardiovascular risks of acrylamide are primarily mediated by its conversion to glycidamide.

Further longitudinal studies are needed to clarify the long-term effects of acrylamide on the cardiovascular system. The findings of this review, which highlight the potential adverse effects of acrylamide exposure, are of high public health relevance, as they provide evidence of associations between key cardiovascular risk factors and disease outcomes. While regulatory measures, such as those implemented in the EU and the US, represent significant progress, additional efforts are needed to increase public awareness of how home-cooking practices contribute to acrylamide exposure. Educating consumers about strategies to minimize acrylamide formation, coupled with continued innovation in food industry practices, will be critical in addressing the broader health risks associated with this substance.

## Figures and Tables

**Figure 1 nutrients-16-04279-f001:**
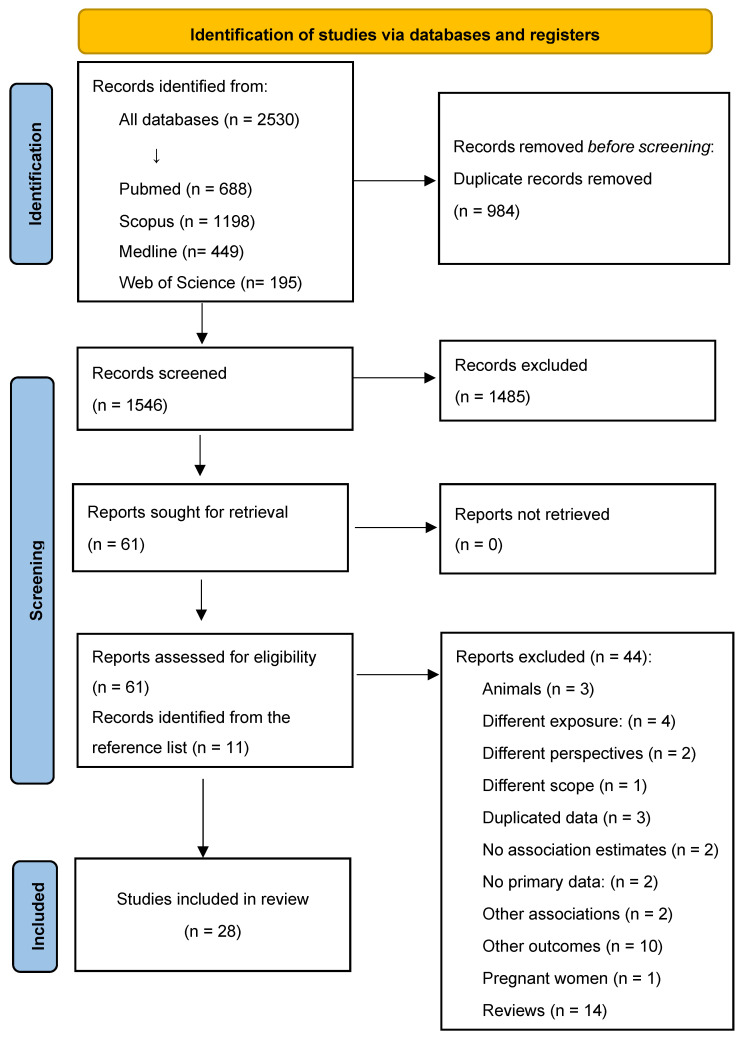
Literature search and study selection.

**Figure 2 nutrients-16-04279-f002:**
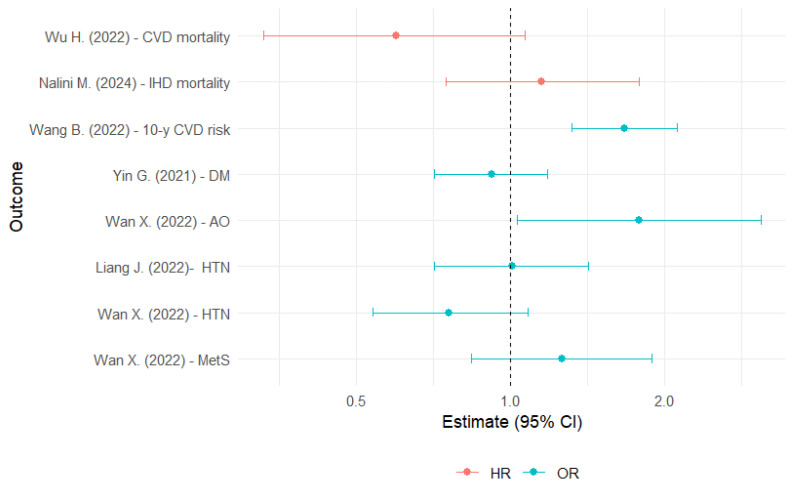
Hazard ratios and odds ratios for the association between glycidamide exposure and cardiovascular risk factors [20,23,25,32,40,43]. AO: abdominal obesity; CI: confidence interval; CVD: cardiovascular disease; DM: diabetes mellitus; IHD: ischemic heart disease; HR: hazard ratio; HTN: hypertension; MetS: metabolic syndrome; OR: odds ratio.

**Figure 3 nutrients-16-04279-f003:**
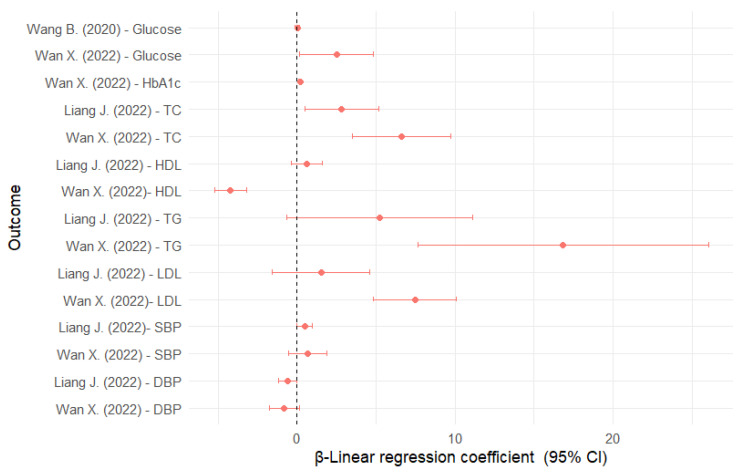
Linear regression coefficients for the association between glycidamide and cardiovascular risk factors [31,40,43]. DBP: diastolic blood pressure; CI: confidence interval; HbA1c: glycated hemoglobin; HDL: high-density lipoprotein; LDL: low-density lipoprotein; SBP: systolic blood pressure; TC: total cholesterol; TG: triglycerides.

**Table 1 nutrients-16-04279-t001:** Characteristics of the included studies.

Author (Year)	Country (Cohort)	Study Design	Sample Size	Age in Years	Men (%)	Acrylamide Exposure (Mean or Median)	Outcomes	Covariates in the Fully Adjusted Model
*CVD mortality*								
Marsh GM et al. (2007) [17]	US and NL	Prospective (US follow-up: 1925–2002; the NL follow-up: 1965–2004)	US: 8508; NL: 344	NA	100	Cumulative acrylamide exposure, mg/m^3^-year (US mean: 0.098; NL mean: 0.114)	-CVD mortality: US: ICD 8th (390–548); NL: ICD 9th (390–548)	Age, duration of employment, race, and time since first employment
Swaen GMH et al. (2007) [18]	US	Retrospective cohort (follow-up: 1955–2001)	696	NA	94.1	Cumulative acrylamide exposure, mg/m^3^ (mean: 4.6)	-Stroke mortality -Heart disease mortality	Age, duration of follow-up, sex, and time interval
Huang M et al. (2018) [19]	US (NHANES 2003–2006)	Prospective (6.7-year follow-up)	5504	≥25	47.9	Blood (pmol/g Hb): -HbAA (median: 61.25) -HbGA (median: 52.50)	-CVD mortality: ICD 10th (I00-I78)	Age, alcohol consumption, BMI, education, energy intake, family history of CVD, PIR, HTN, log-cotinine, log-HbAA, log-HbGA, physical activity, race, and sex
Wu H et al. (2022) [20]	US (NHANES 2003–2006, 2013–2014)	Prospective (follow-up until 2015)	3601	>18	NA	Blood (pmol/g Hb): Diabetics -HbAA (median: 49.7) -HbGA (median: 47.2)	-CVD mortality: ICD 10th (I00–I09, I11, I13, I20–I51, I60–I69)	Age, alcohol consumption, blood pressure, BMI, education, energy intake, income, occupation, physical activity, race, sex, smoking, and taking medication for DM, HTN, or cholesterol
Marques C et al. (2023) [21]	France (E3N French cohort)	Prospective (follow-up: 1993–2014)	72,585	Mean: 52.9	0	Dietary acrylamide intake (µg/day) (mean: 32.6)	-CVD mortality: ICD 9th (390–459) and ICD 10th (I00–I99)	Age, alcohol consumption, birth cohort, BMI, education, energy intake, lipids consumption, menopausal status, physical activity, and smoking
Feng X et al. (2024) [22]	US (NHANES 2005–2006, 2011–2018)	Prospective (6.2-year follow-up)	8799	≥20	49.2	Urine (μg/g Cr): AAMA (median: 54.6)	-CVD mortality: ICD 10th (I00–I09, I11, I13, I20–I51, and I60–I69	Age, alcohol consumption, BMI, education, NHANES cycles, PIR, physical activity, race, sex, and smoking
Nalini M et al. (2024) [23]	Iran (Golestan Cohort Study)	Prospective (8.4-year follow-up)	1176	40–75	62.0	Urine (µg/g creatinine): AAMA (median: 152.2) GAMA (median: 18.8)	-IHD mortality: ICD 10th (I20–I25)	Age, BMI, cotinine, education, enrolment year, marital status, nass use, physical activity, race, and wealth score
*CVD risk*								
Zhang Y et al. (2018) [24]	US (NHANES 2003–2006)	Cross-sectional	8290	≥20	48.2	Blood (pmol/g Hb): -HbAA (mean: 60.70) -HbGA (mean: 51.58)	-CVD (congestive heart failure, coronary heart disease, angina pectoris, heart attack, stroke)	Age, alcohol consumption, BMI, caffeine intake, C-reactive protein, energy intake, family history of CVD, homocysteine, HTN, log-TC, log-cotinine, log-HbAA, log-HBbGA, race, sex, and trans-fat intake
Wang B et al. (2022) [25]	China (Wuhan-Zhuhai cohort)	Cross-sectional	3024	Mean: 53.82	29.4	Urine (µg/mmol Creatinine): -AAMA (median: 3.29) -GAMA (median: 0.50)	-10-year CVD risk	Alcohol consumption, BMI, education, income, LDL, passive smoking, physical activity, region, and sex
Wang X et al. (2023) [26]	US (NHANES 2011–2018)	Cross-sectional	6814	≥20	49.4	Urine (ng/mL): AAMA (mean: 1.70)	-CVD (congestive heart failure, coronary heart disease, angina, heart attack)	Age, cotinine, education, DBP, DM, HDL, HTN, LDL, PIR, race, second-hand smoke, sex, and volatile toxicant questionnaire scores
Ma M et al. (2023) [27]	US (NHANES 2011–2016)	Cross-sectional	5213	Median: 47	48.8	Urine AAMA	-CVD	Age, alcohol consumption, BMI, DM, education, energy intake, eGFR, HDL, HTN, poverty, race, sedentary time, sex, smoking, and TC
Han S et al. (2024) [28]	US (NHANES 2005–2006, 2011–2018)	Cross-sectional	9119	Mean: 66.1 (CVD)	58.6 (CVD)	Urine (ng/mL): AAMA (median: 51.5)	-CVD (congestive heart failure, coronary heart disease, angina, heart attack, or stroke)	Age, alcohol consumption, BMI, DM, education, HTN, PIR, physical activity, race, sex, smoking, TC, urinary albumin, and creatinine
*Diabetes and glucose metabolism*								
Lin CY et al. (2009) [29]	US (NHANES 2003–2004)	Cross-sectional	1356	≥20	48.7	Blood (pmol/g Hb): -HbAA (median: 61.6) -HbGA (median: 57.4)	-Glucose (mmol/L) -HbA1c -Insulin (pmol/L) -HOMA-IR -TC -HDL -TG -WC (cm)	Age, alcohol consumption, BMI, education, HbA1c/insulin/glucose/HOMA-IR, income, race, sex, and smoking
Lin CY et al. (2013) [30]	Taiwan	Cross-sectional	800	Mean 21.3	37.3	Urine (µg/g Creatinine): -AAMA (mean: 52.8)	-Glucose (mg/dL) -Insulin (pM) -HOMA-IR -TC (mg/dL) -HDL (mg/dL) -LDL (mg/dL) -BMI (kg/m^2^)	Age, alcohol consumption, BMI, sex, and smoking
Wang B et al. (2020) [31]	China (Wuhan-Zhuhai cohort)	Cross-sectional	3270	Mean: 53.0	29.9	Urine (µg/mmol Creatinine) -AAMA (median: 3.28) -GAMA (median: 0.50)	-Glucose (mmol/L)	Age, alcohol consumption, BMI, education, family history of DM, income, mean arterial pressure, physical activity, sex, smoking, TC, and TG
Yin G et al. (2021) [32]	US (NHANES 2005–2006, 2013–2016)	Cross-sectional	3577	Mean: 51.4	52.4	Blood (pmol/g Hb): -HbAA (median: 50.4) -HbGA (median: 43.0)	-DM: self-reported diabetes or HbA1c ≥ 6.5% or fasting plasma glucose ≥ 126 mg/dL or 2 h plasma glucose after OGTT ≥ 200 mg/dL	Age, alcohol consumption, BMI, education, HTN, race, sex, and smoking
Hosseini-Esfahani F et al. (2023) [33]	Iran (Tehran lipid and glucose study)	Prospective (6.6-year follow-up)	6022	Mean: 41.5 (men)	45.0	Dietary acrylamide intake (µg/day) (mean: 57.0)	-DM: fasting plasma glucose ≥ 126 mg/dL or 2 h plasma glucose ≥ 200 mg/dL or taking medication for diabetes	Age, BMI, education, energy intake family history of diabetes, HDL, physical activity, sex, smoking, and TG
*Dyslipidemia and lipid metabolism*								
Cheang I et al. (2020) [34]	US (NHANES 2013–2016)	Cross-sectional	2899	Mean: 45.1	50.1	Blood (pmol/g Hb): -HbAA (median: 49.2) -HbGA (median: 42.2)	-TC (mg/dL) -HDL (mg/dL) -TG (mg/dL) -LDL (mg/dL)	Age, alcohol consumption, BMI, DM, education, energy intake, PIR, HTN, physical activity, race, sex, and smoking
Chen WY et al. (2023) [35]	US (NHANES 2011–2015)	Cross-sectional	1410	Mean: 48.2	51.8	Urine (ng/mL): AAMA	-TC (mg/dL) -HDL (mg/dL) -TG (mg/dL) -LDL (mg/dL)	Age, BMI, creatinine, education, energy intake, marital status, PIR, physical activity, race, sex, and smoking
*Obesity and body composition*								
Chu PL et al. (2017) [36]	US (NHANES 2003–2004)	Cross-sectional	3623	≥20	48.4	Blood (pmol/g Hb): -HbAA (median: 60.5) -HbGA (median: 55.6)	-BMI (kg/m^2^) -WC (cm)	Age, caffeine intake, education, energy intake, income, metabolic equivalent intensity level for activity, protein intake, race, saturated fatty acids intake, sex, smoking, and sugar intake
Huang M et al. (2018) [37]	US (NHANES 2003–2006)	Cross-sectional	8364	≥20	48.2	Blood (pmol/g Hb): -HbAA (median: 53.0) -HbGA (median: 50.9)	-GO: BMI ≥ 30 kg/m^2^ -AO: WC >102 cm (men) and >88 cm (women)	Age, alcohol consumption, education, energy intake, PIR, HTN, marital status, physical activity, race, sex, and smoking
Yin T et al. (2022) [38]	US (NHANES 2003–2006, 2013–2014)	Cross-sectional	10,377	Mean: 46.8	51.2	Blood (pmol/g Hb): -HbAA (median: 54.2) -HbGA (median: 49.5)	-GO: BMI ≥ 30 kg/m^2^ -AO: WC >102 cm (men) and >88 cm (women)	Age, alcohol consumption, DM, education, energy intake, HTN, race, sex, and smoking
Lei T et al. (2023) [39]	US (NHANES 2005–2006, 2011–2020)	Cross-sectional	17,524	Mean: 45.09 (obesity)	44.9 (obesity)	Urine (ng/mL): -AAMA (median: 28.7)	-GO: BMI ≥ 30 kg/m^2^ -AO: WC > 102 cm (men) and >88 cm (women)	Age, alcohol consumption, DM, education, energy intake, HTN, marital status, physical activity, race, sex, smoking, urinary albumin, and creatinine
*Hypertension and blood pressure*								
Liang J et al. (2022) [40]	US (NHANES 2003–2006, 2013–2016)	Cross-sectional	3981	Mean: 16.0	51.3	Blood (pmol/g Hb): -HbAA (mean: 65.0) -HbGA (mean: 57.5)	- HTN: blood pressure ≥ 130/80 mmHg or taking antihypertensive treatment -TC (mg/dL) -TG (mg/dL) -LDL (mg/dL) -SBP (mmHg) -DBP (mmHg)	BMI, cotinine, dietary intake of calcium, sodium, and potassium, PIR, gender, and race
McGraw KE et al. (2023) [41]	US (Jackson Heart Study cohort)	Cross-sectional	1194	Mean: 51.4	49.8	Urine (ng/mg creatinine): AAMA (median: 53.7)	-HTA: blood pressure ≥ 140/90 mmHg or taking antihypertensive treatment -SBP -DBP	Age, ambient PM2.5 levels, BMI, DM, education, eGFR, HDL, physical activity, taking medication for HTN, and TG
*Metabolic syndrome*								
Hung CC et al. (2021) [42]	US (NHANES 2003–2006)	Cross-sectional	4813	≥18	50.5 (non-MetS); 49.7 (MetS)	Blood (pmol/g Hb): -HbAA (mean: 49.8) -HbGA (mean: 50.0)	-MetS: Clinical definition by the NCEP-ATP III (3 or more criteria) * -High glucose -Low HDL -High TG -AO -HTN	Age, angina/angina pectoris, aspartate aminotransferase, creatinine, emphysema, race, sex, and smoking
Wan X et al. (2022) [43]	US (NHANES 2003–2006, 2013–2016)	Cross-sectional	4790	≥20	49.6	Blood (pmol/g Hb): -HbAA (mean: 47.7) -HbGA (mean: 60.4)	-MetS: Clinical definition by the NCEP-ATP III (3 or more criteria) * -Glucose -HbA1c -TC -HDL -TG -LDL -AO -HTN -SBP -DBP	Age, alcohol consumption, BMI, caffeine intake, education, energy, intake, log-HbAA or log-HbGA, physical activity, PIR, race, sex, and smoking
Tan L et al. (2024) [44]	US (NHANES 2005–2006, 2011–2020)	Cross-sectional	8223	≥16	50.5	Urine (ng/mL): AAMA (median: 54)	-MetS: Clinical definition by the NCEP-ATP III (3 or more criteria) * -High glucose -Low HDL -High TG -AO -HTN	Age, alcohol consumption, education, marital status, PIR, race, sex, smoking, urinary albumin, and creatinine

AAMA: N-acetyl-S-(2-carbamoylethyl)-l-cysteine); AO: abdominal obesity; BMI: body mass index; CVD: cardiovascular disease; DBP: diastolic blood pressure; DM: diabetes mellitus; eGFR: estimated glomerular filtration rate; GAMA: N-acetyl-S-(2-carbamoyl-2-hydroxyethyl)-l-cysteine; GO: general obesity; HbA1c: glycated hemoglobin; HbAA: hemoglobin adducts of acrylamide; HbGA: hemoglobin adducts of glycidamide; HDL: high-density lipoprotein; HOMA-IR: homeostasis model assessment-insulin resistance; HTN: hypertension; ICD: International Classification of Diseases; IHD: ischemic heart disease; LDL: low-density lipoprotein; MetS: metabolic syndrome; NA: not available; NCEP/ATP III: National Cholesterol Education Program’s Adult Treatment Panel III; NL: the Netherlands; OGTT: oral glucose tolerance test; PIR: poverty income ratio; SBP: systolic blood pressure; TC: total cholesterol; TG: triglycerides; US: United States; WC: waist circumference. * NCEP/ATP III criteria unless otherwise stated: waist circumference ≥102 cm (men) and ≥88 cm (women); blood pressure ≥ 130/85 mmHg or treatment for hypertension; triglycerides level ≥150 mg/dL; HDL < 40 mg/dL (men) and <50 mg/dL (women); fasting blood glucose ≥ 100 mg/dL or treatment for diabetes.

**Table 2 nutrients-16-04279-t002:** Association between acrylamide exposure and cardiovascular disease.

Study	Country (Sample)	Acrylamide Metabolite	Unit of Comparison	Outcome						Results
				CVD Mortality	Stroke Mortality	Heart Disease Mortality	IHD Mortality	CVD	10-y CVD Risk	
Marsh GM et al. (2007) [17]	US and NL	Cumulative acrylamide exposure	Observed mortality vs. expected mortality for the US or NL	↓						CVD mortality (US): SMR = 0.88 (0.84–0.92), *p*-value: < 0.01
				NSA						CVD morality (NL): SMR = 0.73 (0.51–1.02)
Swaen GMH et al. (2007) [18]	US	Cumulative acrylamide exposure	Observed mortality vs. expected mortality for the US		NSA	NSA				Stroke mortality: SMR = 65.3 (21.2–152.30) Heart disease mortality: SMR = 91.8 (68.9–119.7)
Wu H et al. (2022) [20]	US (NHANES 2003–2006, 2013–2014)	HbAA	Q4vsQ1	↑						CVD mortality: HR = 1.84 (1.00–3.37), *p*-trend: 0.002
		HbGA	Q4vsQ1	NSA						CVD mortality: HR = 0.60 (0.33–1.07), *p*-trend: 0.448
		HbAA/HbGA	Q4vsQ1	↑						CVD mortality: HR = 1.61 (1.09–2.39), *p*-trend: 0.062
Marques C et al. (2023) [21]	France (E3N French cohort)	Dietary acrylamide intake	Q4vsQ1	↑						CVD mortality: HR = 1.33 (1.09–1.63), *p*-trend: 0.003
Feng X et al. (2024) [22]	US (NHANES 2005–2006, 2011–2018)	AAMA	1-unit in log_2_ AAMA	NSA						CVD mortality: HR = 1.09 (0.95–1.25)
Nalini M et al. (2024) [23]	Iran (Golestan Cohort Study)	AAMA	T3vsT1				↑			IHD mortality: HR = 1.68 (1.05, 2.69), *p*-trend: 0.025
		GAMA	T3vsT1				NSA			IHD mortality: HR = 1.15 (0.75, 1.78), *p*-trend: 0.522
*CVD risk*										
Wang B et al. (2022) [25]	China (Wuhan-Zhuhai cohort)	AAMA	Q4vsQ1						↑	10 y CVD risk: OR = 1.47 (1.16–1.88), *p*-trend: 0.004
		GAMA	Q4vsQ1						↑	10 y CVD risk: OR = 1.67 (1.32–2.11), *p*-trend: <0.001
		ΣUAAM	Q4vsQ1						↑	10 y CVD risk: OR = 1.51 (1.19–1.91), *p*-trend: 0.001
		GAMA/AAMA	Q4vsQ1						↑	10 y CVD risk: OR = 1.42 (1.10–1.83), *p*-trend: 0.002
Wang X et al. (2023) [26]	US (NHANES 2011–2018)	AAMA	1-unit increase in AAMA					NSA		CVD: OR = 1.1 (0.85–1.43), *p*-value: 0.46
Han S et al. (2024) [28]	US (NHANES 2005–2006, 2011–2018)	AAMA	Q4vsQ1					↑		CVD: OR = 1.54 (1.01–2.35), *p*-trend: 0.021
Ma M et al. (2023) [27]	US (NHANES 2011–2016)	AAMA	Q4vsQ1					↑		CVD: OR = 1.95 (1.09–3.51), *p*-trend: 0.020

AAMA: N-acetyl-S-(2-carbamoylethyl)-l-cysteine); CVD: cardiovascular disease; GAMA: N-acetyl-S-(2-carbamoyl-2-hydroxyethyl)-l-cysteine; HbAA: hemoglobin adducts of acrylamide; HbGA: hemoglobin adducts of glycidamide; HR: hazard ratio; IHD: ischemic heart disease; NSA: not significant association; OR: odds ratio; US: United States. Rows in gray indicate overlapping populations.

**Table 3 nutrients-16-04279-t003:** Association between acrylamide exposure and diabetes and glucose metabolism.

Study	Sample	Acrylamide Metabolite	Unit of Comparison	Outcome						Results
				Glucose	HbA1c	Insulin	HOMA-IR	DM	High Glucose	
Lin CY et al. (2009) [29]	US (NHANES 2003–2004)	HbAA	1-unit increase in natural log HbAA	NSA	NSA	↓	↓			Glucose: β = −0.09 (−0.25, −0.07), *p*-value: 0.262 Log-HbA1c: β = 0.01 (−0.10, 0.03), *p*-value: 0.253 Log-insulin: β = −0.20 (−0.30, −0.10), *p*-value: 0.001 Log-HOMA-IR: β = −0.23 (−0.33, −0.13), *p*-value: <0.001
Lin CY et al. (2013) [30]	Taiwan	AAMA	1-unit increase in natural log AAMA	NSA		NSA	NSA			Glucose: β = −0.192 (−1.60, 1.22), *p*-value: 0.789 Log-insulin: β = 0.023 (−0.04, 0.09), *p*-value: 0.473 Log-HOMA-IR: β = 0.023 (−0.04, 0.09), *p*-value: 0.492
Wang B et al. (2020) [31]	China (Wuhan-Zhuhai cohort)	AAMA	Q4vsQ1	↑						Glucose: β = 0.20 (0.05, 0.35), *p*-trend: 0.008
		GAMA	Q4vsQ1	NSA						Glucose: β = 0.05 (−0.10, 0.19), *p*-trend: 0.581
		ΣUAAM	Q4vsQ1	↑						Glucose: β = 0.17 (0.02, 0.32), *p*-trend: 0.014
Yin G et al. (2021) [32]	US (NHANES 2005–2006, 2013–2016)	HbAA	Q4vsQ1					↓		DM: OR = 0.71 (0.55–0.93), *p*-trend: 0.013
		HbGA	Q4vsQ1					NSA		DM: OR = 0.92 (0.71–1.18), *p*-trend: 0.859
		HbAA + HbGA	Q4vsQ1					NSA		DM: OR = 0.80 (0.62–1.03), *p*-trend: 0.194
		HbGA/HbAA	Q4vsQ1					↑		DM: OR = 1.95 (1.51–2.51), *p*-trend: <0.001
Hosseini-Esfahani F et al. (2023) [33]	Iran (Tehran lipid and glucose study)	Acrylamide intake	Q4vsQ1					NSA		DM: HR = 1.06 (0.98–1.16), *p*-trend:0.13
Hung CC et al. (2021) [42]	US (NHANES 2003–2006)	HbAA	1-unit increase in natural log HbAA						NSA	High glucose: Graphically reported
		HbGA	1-unit increase in natural log HbGA						↓	High glucose: Graphically reported
Wan X et al. (2022) [43]	US (NHANES 2003–2006, 2013–2016)	HbAA	1-unit increase in natural log HbAA	↓	↓					Glucose: β = −5.13 (−7.89, −2.37), *p*-value: < 0.001 HbA1c: β = −0.18 (−0.26, −0.10), *p*-value: < 0.001
		HbGA	1-unit increase in natural log HbGA	↑	↑					Glucose: β = 2.51 (0.16, 4.86), *p*-value: 0.036 HbA1c: β = 0.19 (0.12, 0.26), *p*-value: <0.001
		HbAA + HbGA	1-unit increase in natural log HbAA + HbGA	↓	NSA					Glucose: β = −2.46 (−4.46, −0.46), *p*-value: 0.016 HbA1c: β = 0.02 (−0.04, 0.08), *p*-value: 0.447
		HbGA/HbAA	1-unit increase in natural log HbGA/HbAA	↑	↑					Glucose: β = 3.12 (0.82, 5.42), *p*-value: 0.008 HbA1c: β = 0.19 (0.12, 0.26), *p*-value: <0.001
Tan L et al. (2024) [44]	US (NHANES 2005–2006, 2011–2020)	AAMA	Q4vsQ1						NSA	High glucose: OR = 1.13 (0.93, 1.37), *p*-trend: 0.628

AAMA: N-acetyl-S-(2-carbamoylethyl)-l-cysteine); DM: diabetes mellitus; GAMA: N-acetyl-S-(2-carbamoyl−2-hydroxyethyl)-l-cysteine; HbA1c: glycated hemoglobin; HbAA: hemoglobin adducts of acrylamide; HbGA: hemoglobin adducts of glycidamide; HOMA-IR: homeostasis model assessment-insulin resistance; HR: hazard ratio; NSA: not significant association; OR: odds ratio; US: United States. Rows in gray indicate overlapping populations.

**Table 4 nutrients-16-04279-t004:** Association between acrylamide exposure and dyslipidemia and lipid metabolism.

Study	Country (Sample)	Acrylamide Metabolite	Unit of Comparison	Outcome						Results
				TC	HDL	TG	LDL	Low HDL	High TG	
Lin CY et al. (2009) [29]	US (NHANES 2003–2004)	HbAA	1-unit increase in natural log HbAA	NSA	NSA	NSA				TC: β = 0.02 (−0.14, 0.18), *p*-value: 0.839 HDL: β = −0.05 (−0.11, 0.01), *p*-value: 0.072 Log-TG: β = −0.01 (−0.07, 0.05), *p*-value: 0.742
Lin CY et al. (2013) [30]	Taiwan	AAMA	1-unit increase in natural log AAMA	NSA	NSA		NSA			TC: β = −1.85 (−4.31, 0.61), *p*-value: 0.141 HDL: β = −0.10 (−0.76, 0.57), *p*-value: 0.770 LDL: β = 0.28 (−1.95, 2.51), *p*-value: 0.804
Cheang I et al. (2020) [34]	US (NHANES 2013–2016)	HbAA	Q4vsQ1	NSA	NSA	↑	NSA			TC: β = 1.74 (−3.22, 6.70), *p*-trend: 0.606 HDL: β = −0.32 (−2.17, 1.53), *p*-trend: 0.718 TG: β = 23.63 (6.01, 41.24), *p*-trend: 0.047 LDL: β = 1.12 (−5.59, 7.82), *p*-trend: 0.909
		HbGA	Q4vsQ1	↑	↓	↑	NSA			TC: β= 6.78 (2.17, 11.40), *p*-trend: 0.002 HDL: β= −5.12 (−6.84, −3.40), *p*-trend: < 0.001 TG: β= 25.32 (8.98, 41.65), *p*-trend: 0.004 LDL: β= 5.89 (−0.32, 12.10), *p*-trend: 0.026
		HbAA + HbGA	Q4vsQ1	NSA	↓	↑	NSA			TC: β= 3.81 (−1.02, 8.63), *p*-trend: 0.106 HDL: β= −2.09 (−3.89, −0.29), *p*-trend: 0.049 TG: β= 25.65 (8.63, 42.68), *p*-trend: 0.017 LDL: β= 1.87 (−4.59, 8.34), *p*-trend: 0.443
		HbGA/HbAA	Q4vsQ1	↑	↓	↑	↑			TC: β= 12.79 (8.18, 17.40), *p*-trend: <0.001 HDL: β= −7.24 (−8.94, −5.53), *p*-trend: <0.001 TG: β= 27.33 (10.72, 43.94), *p*-trend: 0.002 LDL: β= 11.10 (4.82,17.39), *p*-trend: 0.001
Liang J et al. (2022) [40]	US (NHANES 2003–2006, 2013–2016)	HbGA	1-unit increase in Natural log HbGA	↑	NSA	NSA	NSA			TC: β= 2.83 (0.49, 5.18), *p*-value: 0.018 HDL: β= 0.65 (−0.33, 1.63), *p*-value: 0.192 TG: β= 5.23 (−0.65, 11.11), *p*-value: 0.081 LDL: β= 1.52 (−1.59, 4.63), *p*-value: 0.337
Hung CC et al. (2021) [42]	US (NHANES 2003–2006)	HbAA	1-unit increase in natural log HbAA					↓	↓	Low HDL and High TG: Graphically reported
		HbGA	1-unit increase in natural log HbGA					NSA	NSA	Low HDL and High TG: Graphically reported
Wan X et al. (2022) [43]	US (NHANES 2003–2006, 2013–2016)	HbAA	1-unit increase in natural log HbAA	↓	↑	↓	↓			TC: β= −4.62 (−8.28, −0.96), *p*-value: 0.013 HDL: β= 3.89 (2.67, 5.10), *p*-value: <0.001 TG: β= −18.85 (−29.69, −8.00), *p*-value: 0.001 LDL: β= −5.41 (−8.49, −2.33), *p*-value: 0.001
		HbGA	1-unit increase in natural log HbGA	↑	↓	↑	↑			TC: β= 6.60 (3.49, 9.72), *p*-value: <0.001 HDL: β= −4.20 (−5.23, −3.17), *p*-value: <0.001 TG: β= 16.86 (7.64, 26.08), *p*-value: <0.001 LDL: β= 7.47 (4.85, 10.08), *p*-value: <0.001
		HbAA + HbGA	1-unit increase in natural log HbAA + HbGA	↑	NSA	NSA	↑			TC: β= 3.27 (0.61, 5.92), *p*-value: 0.016 HDL: β= −0.59 (−1.48, 0.29), *p*-value: 0.191 TG: β= 1.56 (−6.31, 9.42), *p*-value: 0.698 LDL: β= 2.98 (0.74, 5.21), *p*-value: 0.009
		HbGA/HbAA	1-unit increase in natural log HbGA/HbAA	↑	↓	↑	↑			TC: β= 6.14 (3.09, 9.19), *p*-value: <0.001 HDL: β= −4.13 (−5.14, −3.11), *p*-value: <0.001 TG: 17.32 (8.28, 26.37), *p*-value: <0.001 LDL: β= 6.99 (4.42, 9.55), *p*-value: <0.001
Chen WY et al. (2023) [35]	US (NHANES 2011–2015)	AAMA	1-unit increase in natural log AAMA	NSA	↑	NSA	NSA			TC: β= −0.63 (−4.46, 3.2), *p*-value: 0.75 ln-HDL: 1.28 (0.21, 2.36), *p*-value: 0.03 ln-TG: β= −0.09 (−6.82, 6.64), *p*-value: 0.98 LDL: −1.87 (−5.33,1.58), *p*-value: 0.3
Tan L et al. (2024) [44]	US (NHANES 2005–2006, 2011–2020)	AAMA	Q4vsQ1					NSA	NSA	Low HDL: OR= 0.89 (0.73, 1.07), *p*-trend: 0.137 High TG: OR= 0.90 (0.75, 1.08), *p*-trend: 0.097

AAMA: N-acetyl-S-(2-carbamoylethyl)-l-cysteine); GAMA: N-acetyl-S-(2-carbamoyl-2-hydroxyethyl)-l-cysteine; HbAA: hemoglobin adducts of acrylamide; HbGA: hemoglobin adducts of glycidamide; HDL: high-density lipoprotein; LDL: low-density lipoprotein; ln: natural log; NSA: not significant association; OR: odds ratio; TC: total cholesterol; TG: triglycerides; US: United States. Rows in gray indicate overlapping populations.

**Table 5 nutrients-16-04279-t005:** Association between acrylamide exposure and obesity and body composition.

Study	Country (Sample)	Acrylamide Metabolite	Unit of Comparison	Outcome				Results
				BMI	WC	GO	AO	
Lin CY et al. (2009) [29]	US (NHANES 2003–2004)	HbAA	1-unit increase in natural log HbAA		NSA			WC: β = 0.44 (−1.74, 2.62), *p*-value: 0.697
Lin CY et al. (2013) [30]	Taiwan	AAMA	1-unit increase in natural log AAMA	NSA				BMI: β = −0.094 (−0.39, 0.20), *p*-value: 0.535
Chu PL et al. (2017) [36]	US (NHANES 2003–2004)	HbAA	1-unit increase in natural log HbAA	↓	↓			BMI: β = −1.46 (−2.07, −0.85), *p*-value: <0.001 WC: β = −3.72 (−5.33, −2.11), *p*-value: <0.001
		HbGA	1-unit increase in natural log HbGA	NSA				BMI: β = 0.38 (−0.19, 0.95), *p*-value: 0.217 WC: β = 0.58 (−0.81, 1.97), *p*-value: 0.429
Huang M et al. (2018) [37]	US (NHANES 2003–2006)	HbAA	Q4vsQ1			↓	↓	GO: OR = 0.67 (0.55–0.81), *p*-trend: <0.0001 AO: OR = 0.66 (0.57–0.75), *p*-trend: <0.0001
		HbGA	Q4vsQ1			↑	↑	GO: OR = 1.40 (1.17–1.68), *p*-trend: 0.0004 AO: OR = 1.43 (1.19–1.72), *p*-trend: 0.0008
		HbAA + HbGA	Q4vsQ1			NSA	NSA	GO: OR = 0.90 (0.73–1.13), *p*-trend: 0.4877 AO: OR = 0.88 (0.74–1.05), *p*-trend: 0.4151
		HbGA/HbAA	Q4vsQ1			↑	↑	GO: OR = 2.86 (2.43–3.38), *p*-trend: <0.0001 AO: OR = 2.91 (2.40–3.54), *p*-trend: <0.0001
Yin T et al. (2022) [38]	US (NHANES 2003–2006, 2013–2014)	HbAA	1-unit in log2 HbAA			↓	↓	GO: OR = 0.80 (0.76–0.85), *p*-value: <0.001 AO: OR= 0.79 (0.75–0.83), *p*-value: <0.001
		HbGA	1-unit in log2 HbGA			NSA	NSA	GO: OR = 1.03 (0.98–1.08) AO: OR = 1.03 (0.98–1.08)
		HbAA + HbGA	1-unit in log2 HbAA + HbGA			↓	↓	GO: OR = 0.88 (0.83–0.93), *p*-value: <0.001 AO: OR = 0.86 (0.82–0.91), *p*-value: <0.001
Lei T et al. (2023) [39]	US (NHANES 2005–2006, 2011–2020)	AAMA	Q4vsQ1			↓	↓	GO: OR = 0.79 (0.67–0.92), *p*-value: <0.01 AO: OR = 0.79 (0.65–0.90), *p*-value: <0.001
Hung CC et al. (2021) [42]	US (NHANES 2003–2006)	HbAA	1-unit increase in natural log HbAA				NSA	AO: Graphically reported
		HbGA	1-unit increase in natural log HbGA				NSA	AO: Graphically reported
Wan X et al. (2022) [43]	US (NHANES 2003–2006, 2013–2016)	HbAA	Q4vsQ1				↓	AO: OR = 0.36 (0.23–0.58), *p*-trend: <0.001
		HbGA	Q4vsQ1				↑	AO: OR = 1.78 (1.03–3.08), *p*-trend: 0.192
		HbAA + HbGA	Q4vsQ1				NSA	AO: OR = 0.70 (0.47–1.03), *p*-trend: 0.051
		HbGA/HbAA	Q4vsQ1				↑	AO: OR = 1.95 (1.28–2.99), *p*-trend: 0.010
Tan L et al. (2024) [44]	US (NHANES 2005–2006, 2011–2020)	AAMA	Q4vsQ1				NSA	AO: OR = 1.01 (0.84, 1.21), *p*-trend: 0.289

AAMA: N-acetyl-S-(2-carbamoylethyl)-l-cysteine); AO: abdominal obesity; BMI: body mass index; GAMA: N-acetyl-S-(2-carbamoyl-2-hydroxyethyl)-l-cysteine; GO: general obesity; HbAA: hemoglobin adducts of acrylamide; HbGA: hemoglobin adducts of glycidamide; NSA: not significant association; OR: odds ratio; US: United State; WC: waist circumference. Rows in gray indicate overlapping populations.

**Table 6 nutrients-16-04279-t006:** Association between acrylamide exposure and hypertension and blood pressure.

Study	Country (Sample)	Acrylamide Metabolite	Unit of Comparison	Outcome			Results
				SBP	DBP	HTN	
Liang J et al. (2022) [40]	US (NHANES 2003–2006, 2013–2016)	HbAA	Q4vsQ1			NSA	HTN: OR = 1.01 (0.72–1.42), *p*-trend: 0.794
		HbGA	Q4vsQ1			NSA	HTN: OR = 1.01 (0.71–1.42), *p*-trend: 0.795
		HbAA	1-unit increase in natural log HbAA	NSA	NSA		ln-SBP: β = 0.27 (−0.37, 0.90), *p*-value: 0.413 ln-DBP: β = −0.39 (−1.18, 0.39), *p*-value: 0.325
		HbGA	1-unit increase in natural log HbGA	↑	NSA		ln-SBP: β = 0.49 (0, 0.97), *p*-value: 0.048 ln-DBP: β = −0.59 (−1.19, 0), *p*-value: 0.051
Hung CC et al. (2021) [42]	US (NHANES 2003–2006)	HbAA	1-unit increase in natural log HbAA			NSA	HTN: Graphically reported
		HbGA	1-unit increase in natural log HbGA			NSA	HTN: Graphically reported
Wan X et al. (2022) [43]	US (NHANES 2003–2006, 2013–2016)	HbAA	Q4vsQ1			NSA	HTN: OR = 0.70 (0.47–1.03), *p*-trend: 0.046
		HbGA	Q4vsQ1			NSA	HTN: OR = 0.76 (0.54–1.08), *p*-trend: 0.129
		HbAA + HbGA	Q4vsQ1			NSA	HTN: OR = 0.83 (0.62–1.11), *p*-trend: 0.847
		HbGA/HbAA	Q4vsQ1			NSA	HTN: OR = 0.99 (0.74–1.33), *p*-trend: 0.905
		HbAA	1-unit increase in natural log HbAA	NSA	NSA		SBP: β = 0.58 (−0.84, 1.99), *p*-value: 0.425 DBP: β= 0.49 (−0.62, 1.60), *p*-value: 0.385
		HbGA	1-unit increase in natural log HbGA	NSA	NSA		SBP: β = 0.67 (−0.53, 1.88), *p*-value: 0.272 DBP: β = −0.81 (−1.75,0.14), *p*-value: 0.094
		HbAA + HbGA	1-unit increase in natural log HbAA + HbGA	↑	NSA		SBP: β = 1.29 (0.26, 2.31), *p*-value: 0.014 DBP: β = −0.46 (−1.26, 0.35), *p*-value: 0.267
		HbGA/HbAA	1-unit increase in natural log HbGA/HbAA	NSA	NSA		SBP: β = 0.38 (−0.80, 1.56), *p*-value: 0.525 DBP: β = −0.73 (−1.30, 0.55), *p*-value: 0.120
McGraw KE et al. (2023) [41]	US (Jackson Heart Study cohort)	AAMA	Per IQR of AAMA	NSA	NSA	NSA	HTN: RR = 1.02 (0.97–1.08), *p*-value: 0.38 SBP β = 0.24 (−1.20, 1.68), *p*-value: 0.75 SBP: DBP: β = −0.07 (−0.81, 0.67), *p*-value: 0.85
Tan L et al. (2024) [44]	US (NHANES 2005–2006, 2011–2020)	AAMA	Q4vsQ1			↓	HTN: OR = 0.60 (0.47–0.76), *p*-trend: 0.003

HbAA: hemoglobin adducts of acrylamide; HbGA: hemoglobin adducts of glycidamide; HTN: hypertension; IQR: interquartile range; NSA: not significant association; OR: odds ratio; RR: relative risk; SBP: systolic blood pressure; US: United States. Rows in gray indicate overlapping populations.

**Table 7 nutrients-16-04279-t007:** Association between acrylamide exposure and metabolic syndrome.

Study	Country (Sample)	Acrylamide Metabolite	Unit of Comparison	MetS	Results
Hung CC et al. (2021) [42]	US (NHANES 2003–2006)	HbAA	Q4vsQ1	↓	MetS: OR = 0.60 (0.40–0.88), *p*-value: 0.009
		HbGA	Q4vsQ1	NSA	MetS: OR = 1.02 (0.72–1.45), *p*-value: 0.911
Wan X et al. (2022) [43]	US (NHANES 2003–2006, 2013–2016)	HbAA	Q4vsQ1	↓	MetS: OR = 0.60 (0.40–0.89), *p*-trend: 0.001
		HbGA	Q4vsQ1	NSA	MetS: OR = 1.26 (0.84–1.89), *p*-trend: 0.232
		HbAA + HbGA	Q4vsQ1	NSA	MetS: OR = 0.93 (0.71–1.21), *p*-trend: 0.371
		HbGA/HbAA	Q4vsQ1	↑	MetS: OR = 1.61 (1.18–2.20), *p*-trend: 0.001
Tan L et al. (2024) [44]	US (NHANES 2005–2006, 2011–2020)	AAMA	Q4vsQ1	↓	MetS: OR = 0.78 (0.64–0.95), *p*-trend: < 0.001

MetS: metabolic syndrome; NSA: not significant association; OR: odds ratio; US: United States. Rows in gray indicate overlapping populations.

**Table 8 nutrients-16-04279-t008:** Main conclusions of the review.

Outcome	Main Conclusion	Study
CVD mortality	HbAA, dietary acrylamide intake, and AAMA are associated with a higher risk of CVD mortality.	Wu H et al. (2022) [20] Marques C et al. (2023) [21] Nalini M et al. (2024) [23]
CVD risk	AAMA is associated with higher 10 y CVD risk and CVD prevalence.	Wang B et al. (2022) [25] Han S et al. (2024) [28]
	GAMA is associated with higher 10 y CVD risk.	Wang B et al. (2022) [25]
Diabetes and glucose metabolism	AAMA is positively associated with glucose levels. HbAA is associated with lower DM prevalence, glucose, and HbA1c levels.	Wang B et al. (2020) [31] Yin G et al. (2021) [32] Wan X et al. (2022) [43]
	HbGA is positively associated with glucose and HbA1c levels.	Wan X et al. (2022) [43]
Dyslipidemia and lipid metabolism	HbAA is negatively associated with TC, TG, and LDL levels. HbAA and AAMA are positively associated with HDL levels.	Wan X et al. (2022) [43] Chen WY et al. (2023) [35]
	HbGA is positively associated with TC, TG, and LDL levels and negatively associated with HDL levels in US adolescents.	Liang J et al. (2022) [40] Wan X et al. (2022) [43]
Obesity and body composition	AAMA is associated with a lower prevalence of GO and AO. AAMA and HbAA are associated with a lower prevalence of AO.	Lei T et al. (2023) [39] Wan X et al. (2022) [43]
	HbGA is associated with a higher prevalence of AO.	Wan X et al. (2022) [43]
Hypertension and blood pressure	AAMA is associated with a lower prevalence of HTN.	Tan L et al. (2024) [44]
	HbGA is associated with higher SBP levels in US adolescents.	Liang J et al. (2022) [40]
Metabolic syndrome	HbAA and AAMA are associated with a lower prevalence of MetS.	Wan X et al. (2022) [43] Tan L et al. (2024) [44]

AAMA: N-acetyl-S-(2-carbamoylethyl)-l-cysteine); CVD: cardiovascular disease; DM: diabetes mellitus; GAMA: N-acetyl-S-(2-carbamoyl-2-hydroxyethyl)-l-cysteine; GO: general obesity; HbA1c: glycated hemoglobin; HbAA: hemoglobin adducts of acrylamide; HbGA: hemoglobin adducts of glycidamide; HTN: hypertension; MetS: metabolic syndrome; US: United States.

## Data Availability

The original contributions presented in the study are included in the article and Appendix A, further inquiries can be directed to the corresponding author.

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
