# Peer review of "Acrylamide Exposure and Cardiovascular Risk: A Systematic Review"

_nutrients, 2024, doi:10.3390/nu16244279_

Round 1
Reviewer 1 Report
Comments and Suggestions for Authors
In order for the authors' otherwise very good synthesis to be more relevant, I suggest that they develop and insert in the paper a logical scheme to highlight the effects of exposure to acrylamide and its harmful effects on the state of health.
I also recommend that the conclusions be more consistent
Author Response
In order for the authors' otherwise very good synthesis to be more relevant, I suggest that they develop and insert in the paper a logical scheme to highlight the effects of exposure to acrylamide and its harmful effects on the state of health. I also recommend that the conclusions be more consistent.
Author response: Thank you for this comment. In order to highlight the results of the review, and according to the comments of another reviewer, we have included a table with the main findings, those that support our conclusion. We have also modified the conclusion to make it clearer. You can find it below and in the new version of the manuscript.
Table 8. Main conclusions of the review.
|
Outcome |
Main conclusion |
Study |
|
CVD mortality |
HbAA, dietary acrylamide intake and AAMA were associated with a higher risk of CVD mortality |
Wu H et al. (2022) [20] Marques C et al. (2023) [21] Nalini M et al. (2024) [23] |
|
|
||
|
CVD risk |
AAMA was associated with a higher 10-y CVD risk and CVD prevalence |
|
|
Wang B et al. (2022) [25] Han S et al. (2024) [28] |
||
|
|
||
|
|
||
|
GAMA is associated with a higher 10-y CVD risk |
Wang B et al. (2022) [25] |
|
|
Diabetes and glucose metabolism |
AAMA was positively associated with glucose levels. HbAA was associated with lower DM prevalence, glucose and HbA1c levels |
Wang B et al. (2020) [31] Yin G et al. (2021) [32] Wan X et al. (2022) [43]
|
|
HbGA was positively associated with glucose and HbA1c levels |
Wan X et al. (2022) [43] |
|
|
Dyslipidemia and lipid metabolism |
HbAA was negatively associated with TC, TG and LDL levels. HbAA and AAMA were positively associated with HDL levels |
Wan X et al. (2022) [43] Chen WY et al. (2023) [35]
|
|
HbGA was positively associated with TC, TG and LDL levels and negatively associated with HDL levels in US adolescents |
Liang J et al. (2022) [40] Wan X et al. (2022) [43]
|
|
|
Obesity and body composition |
AAMA was associated with a lower prevalence of GO and AO. AAMA and HbAA were associated with lower prevalence of AO |
Lei T et al. (2023) [39] Wan X et al. (2022) [43]
|
|
HbGA was associated with a higher prevalence of AO |
Wan X et al. (2022) [43]
|
|
|
Hypertension and blood pressure |
AAMA was associated with a lower prevalence of HTN |
Tan L et al. (2024) [44] |
|
HbGA was associated with higher SBP levels in US adolescents |
Liang J et al. (2022) [40]
|
|
|
Metabolic syndrome |
HbAA and AAMA were associated with lower prevalence of MetS |
Wan X et al. (2022) [43] Tan L et al. (2024) [44] |
AAMA: N-acetyl-S-(2-carbamoylethyl)-l-cysteine); CVD: cardiovascular disease; DM: diabetes mellitus; GAMA: N-acetyl-S-(2-carbamoyl-2-hydroxyethyl)-l-cysteine; GO: general obesity; HbA1c: glycated hemoglobin; HbAA: hemoglobin adducts of acrylamide; HbGA: hemoglobin adducts of glycidamide; HTN: hypertension; MetS: metabolic syndrome; US: United States.
CONCLUSIONS
"Acrylamide exposure is associated with a higher risk of CVD mortality, 10-year CVD risk and CVD prevalence. However, it is inversely associated with the prevalence of DM, GO, AO, HTN and MetS. In addition, acrylamide is negatively associated with glucose, HbA1c and lipid levels (TC, TG and LDL) and positively associated with HDL. In contrast, glycidamide - its most reactive metabolite - is associated with adverse out-comes, including higher 10-year CVD risk, higher prevalence of AO, increased glucose and HbA1c levels, increased lipid levels (TC, TG and LDL), higher SBP and lower HDL levels. These findings suggest that the cardiovascular risks of acrylamide are primarily mediated by its conversion to glycidamide.
Further longitudinal studies are needed to clarify the long-term effects of acrylamide on the cardiovascular system. The findings of this review, which highlight the po-tential adverse effects of acrylamide exposure, are of high public health relevance, as they provide evidence of associations with key cardiovascular risk factors and disease outcomes. While regulatory measures, such as those implemented in the EU and the US, represent significant progress, additional efforts are needed to increase public awareness of how home cooking practices contribute to acrylamide exposure. Educating consumers about strategies to minimize acrylamide formation, coupled with continued innovation in food industry practices, will be critical in addressing the broader health risks associated with this substance”.
Reviewer 2 Report
Comments and Suggestions for Authors
This paper is an excellent review article that comprehensively summarizes the relationship between acrylamide exposure and cardiovascular risk. The data obtained from a literature search of four databases is systematically organized and analyzed, but to make it more useful for readers, please consider the following points.
1. The data is well organized in Tables 1-7 for each reference, but it would be easier to understand if the conclusions drawn from these data were organized and explained in a separate table.
2. Were there any data that contradicted each other in the papers? If there are any points or problems to be aware of among the data presented, they should be stated in a separate section.
3. The characteristics of this review are summarized at the end of the discussion, which is easy to understand, but the impact would be increased if the usefulness of the review were emphasized, such as how and where it is used.
Author Response
This paper is an excellent review article that comprehensively summarizes the relationship between acrylamide exposure and cardiovascular risk. The data obtained from a literature search of four databases is systematically organized and analyzed, but to make it more useful for readers, please consider the following points.
- The data is well organized in Tables 1-7 for each reference, but it would be easier to understand if the conclusions drawn from these data were organized and explained in a separate table.
Author response: Thank you for this comment. In order to better understand the results of our review, we included another table with the main findings, those that support our conclusions. You can find it below and in the new version of the manuscript.
Table 8. Main conclusions of the review.
|
Outcome |
Main conclusion |
Study |
|
CVD mortality |
HbAA, dietary acrylamide intake and AAMA were associated with a higher risk of CVD mortality |
Wu H et al. (2022) [20] Marques C et al. (2023) [21] Nalini M et al. (2024) [23] |
|
|
||
|
CVD risk |
AAMA was associated with a higher 10-y CVD risk and CVD prevalence |
|
|
Wang B et al. (2022) [25] Han S et al. (2024) [28] |
||
|
|
||
|
|
||
|
GAMA is associated with a higher 10-y CVD risk |
Wang B et al. (2022) [25] |
|
|
Diabetes and glucose metabolism |
AAMA was positively associated with glucose levels. HbAA was associated with lower DM prevalence, glucose and HbA1c levels |
Wang B et al. (2020) [31] Yin G et al. (2021) [32] Wan X et al. (2022) [43]
|
|
HbGA was positively associated with glucose and HbA1c levels |
Wan X et al. (2022) [43] |
|
|
Dyslipidemia and lipid metabolism |
HbAA was negatively associated with TC, TG and LDL levels. HbAA and AAMA were positively associated with HDL levels |
Wan X et al. (2022) [43] Chen WY et al. (2023) [35]
|
|
HbGA was positively associated with TC, TG and LDL levels and negatively associated with HDL levels in US adolescents |
Liang J et al. (2022) [40] Wan X et al. (2022) [43]
|
|
|
Obesity and body composition |
AAMA was associated with a lower prevalence of GO and AO. AAMA and HbAA were associated with lower prevalence of AO |
Lei T et al. (2023) [39] Wan X et al. (2022) [43]
|
|
HbGA was associated with a higher prevalence of AO |
Wan X et al. (2022) [43]
|
|
|
Hypertension and blood pressure |
AAMA was associated with a lower prevalence of HTN |
Tan L et al. (2024) [44] |
|
HbGA was associated with higher SBP levels in US adolescents |
Liang J et al. (2022) [40]
|
|
|
Metabolic syndrome |
HbAA and AAMA were associated with lower prevalence of MetS |
Wan X et al. (2022) [43] Tan L et al. (2024) [44] |
AAMA: N-acetyl-S-(2-carbamoylethyl)-l-cysteine); CVD: cardiovascular disease; DM: diabetes mellitus; GAMA: N-acetyl-S-(2-carbamoyl-2-hydroxyethyl)-l-cysteine; GO: general obesity; HbA1c: glycated hemoglobin; HbAA: hemoglobin adducts of acrylamide; HbGA: hemoglobin adducts of glycidamide; HTN: hypertension; MetS: metabolic syndrome; US: United States.
- Were there any data that contradicted each other in the papers? If there are any points or problems to be aware of among the data presented, they should be stated in a separate section.
Author response: Thank you for this comment. The reviewer is right. We believe that this review highlights both consistent results as well as contradictory data that can be partially explained by the measurement of the exposure and by the metabolism of acrylamide itself. We added a paragraph mentioning these contradictory results. You can see it below and in the new version of the manuscript.
“Some outcomes show contradictory results depending on the biomarker used to assess acrylamide exposure. For instance, AAMA was associated with increased glu-cose levels, whereas HbAA was associated with decreased glucose levels. This discrepancy may arise from the fact that HbAA reflects long-term exposure to acrylamide, whereas AAMA reflects short-term exposure [9]. Furthermore, markers of acrylamide and its metabolite glycidamide often show opposite associations for most outcomes. In our results, acrylamide was negatively associated with most cardiovascular risk factors, whereas glycidamide was positively associated with most of them. This highlights the critical role of the conversion of acrylamide to glycidamide, as glycidamide is more toxic and induces greater oxidative damage, which may explain its adverse cardiovascular associations”.
- The characteristics of this review are summarized at the end of the discussion, which is easy to understand, but the impact would be increased if the usefulness of the review were emphasized, such as how and where it is used.
Author response: Thank you for this comment. We modified the conclusion to include some points related to the impact of the results of this review as well as its usefulness. You can see it below and in the new version of the manuscript.
CONCLUSIONS
"Acrylamide exposure is associated with a higher risk of CVD mortality, 10-year CVD risk and CVD prevalence. However, it is inversely associated with the prevalence of DM, GO, AO, HTN and MetS. In addition, acrylamide is negatively associated with glucose, HbA1c and lipid levels (TC, TG and LDL) and positively associated with HDL. In contrast, glycidamide - its most reactive metabolite - is associated with adverse out-comes, including higher 10-year CVD risk, higher prevalence of AO, increased glucose and HbA1c levels, increased lipid levels (TC, TG and LDL), higher SBP and lower HDL levels. These findings suggest that the cardiovascular risks of acrylamide are primarily mediated by its conversion to glycidamide.
Further longitudinal studies are needed to clarify the long-term effects of acrylamide on the cardiovascular system. The findings of this review, which highlight the potential adverse effects of acrylamide exposure, are of high public health relevance, as they provide evidence of associations with key cardiovascular risk factors and disease outcomes. While regulatory measures, such as those implemented in the EU and the US, represent significant progress, additional efforts are needed to increase public awareness of how home cooking practices contribute to acrylamide exposure. Educating consumers about strategies to minimize acrylamide formation, coupled with continued innovation in food industry practices, will be critical in addressing the broader health risks associated with this substance”.
Summary of the changes made to the manuscript
- Revised one paragraph in the introduction section.
- Adjusted the order of measures of effect (OR, HR, and RR).
- Updated the title of Table 1.
- Replaced "Cerebrovascular disease (CeVD)" with "stroke".
- Corrected "Blood" in Table 1.
- Added a paragraph to the discussion section explaining the contradictory results.
- Included Table 8 in the discussion section to summarize the main conclusions of the review.
- Revised the limitations section.
- Updated the conclusions section.
Reviewer 3 Report
Comments and Suggestions for Authors
The paper assesses the correlation between acrylamide exposure and cardiovascular risk by a systematic literature review. Acrylamide, a compound generated during high-temperature cooking, is examined for its direct effects and those of its metabolite, glycidamide, on diverse cardiovascular outcomes. The review aggregates data from many research, mainly cross-sectional, to examine the relationship between acrylamide exposure, its conversion to glycidamide, and cardiovascular mortality, as well as other risk variables like glucose and cholesterol levels.
The authors have comprehensively reviewed and analyzed the data from various databases and presented the data in the paper. However, I have some questions for the authors. Please find my question below-
1. How did you guarantee the comparability of outcome measures across the included studies, considering the heterogeneity in their design and the biomarkers employed for evaluating acrylamide exposure?
2. The study encompasses both cross-sectional and cohort methods; how do the intrinsic constraints of both methodologies influence the interpretation of causation between acrylamide exposure and cardiovascular risks?
3. What measures were implemented to mitigate the heterogeneity across the studies regarding demographic, geographical location, and methodologies for acrylamide measurement in the synthesis of the results?
4. Considering the inverse relationship shown between acrylamide exposure and some cardiovascular risk factors, can there be potential confounding variables that were insufficiently controlled in the studies?
5. The significance of glycidamide in affecting cardiovascular risks is emphasized; what are the metabolic pathway differences between acrylamide and glycidamide that may elucidate their contrasting relationships with cardiovascular health outcomes?
6. In light of the varied outcomes reported in the literature, how do the results of this review correspond with or diverge from current meta-analyses regarding acrylamide exposure and cardiovascular effects?
7. What are the consequences of the substantial dependence on data from the NHANES database, and how could this affect the generalizability of the findings to populations beyond the United States?
8. The review recommends conducting stratified analyses based on smoking status; what is the interaction between smoking and acrylamide metabolism, and what are the consequences for cardiovascular risk?
9. Given that cross-sectional research constitute the predominant evidence base, what particular longitudinal data would be essential to validate these apparent associations?
Author Response
The paper assesses the correlation between acrylamide exposure and cardiovascular risk by a systematic literature review. Acrylamide, a compound generated during high-temperature cooking, is examined for its direct effects and those of its metabolite, glycidamide, on diverse cardiovascular outcomes. The review aggregates data from many research, mainly cross-sectional, to examine the relationship between acrylamide exposure, its conversion to glycidamide, and cardiovascular mortality, as well as other risk variables like glucose and cholesterol levels.
The authors have comprehensively reviewed and analyzed the data from various databases and presented the data in the paper. However, I have some questions for the authors. Please find my question below-
- How did you guarantee the comparability of outcome measures across the included studies, considering the heterogeneity in their design and the biomarkers employed for evaluating acrylamide exposure?
Author response: Thank you for your comment. The comparability of the outcome measures between the included studies cannot be unsured, as presented in the characteristics of the included studies in table 1. In fact, this point prevents us from performing a meta-analysis combining the results. On the other hand, the studies were organized into homogeneous categories, with separate analysis of biomarkers, and when available the use of standardized metrics for reporting findings. However, heterogeneity persists as an important limitation and highlights the need for future studies with uniform methodologies.
We have modified the limitations section to reflect this point:
“There are also some limitations. First, most of the studies included in this review are mainly based on the US NHANES, which may limit the generalizability and applicability of the findings to other populations. Second, although interest in acrylamide exposure has increased in recent years, the current evidence on its associations with CVD is scarce, fragmented, not standardized, and yields mixed results. Also, most of the results were not stratified according to tobacco consumption, that could be an important confounding factor. In addition, it seems that acrylamide and glycidamide are not associated in the same direction with certain cardiovascular risk factors. Thirdly, a meta-analysis could not be performed because of differences in outcome measures, statistical approaches, the partly overlapping populations or the lack of studies. Finally, except for mortality, most of the conclusions are based on a small number of studies and cross-sectional designs that do not allow us to establish causality.”
- The study encompasses both cross-sectional and cohort methods; how do the intrinsic constraints of both methodologies influence the interpretation of causation between acrylamide exposure and cardiovascular risks?
Author response: Thank you for your comment. The reviewer is right in this comment. In this field, and given the potentially deleterious effect of exposure, longitudinal studies provide us with the highest degree of evidence available. This would lead us to emphasize the results related to cardiovascular mortality. For the rest of the end-points, the cross-sectional design prevents us from being able to make causal relationships. Results from cross-sectional studies should be interpreted with caution, always considering potential biases and the need for further well-designed prospective research to corroborate the findings. However, with the available information, the use of acrylamide has already been regulated in industrial food production.
To emphasize the reviewer's comment, we have modified the discussion, you can see it below and in the new version of the manuscript.
CONCLUSIONS
"Acrylamide exposure is associated with a higher risk of CVD mortality, 10-year CVD risk and CVD prevalence. However, it is inversely associated with the prevalence of DM, GO, AO, HTN and MetS. In addition, acrylamide is negatively associated with glucose, HbA1c and lipid levels (TC, TG and LDL) and positively associated with HDL. In contrast, glycidamide - its most reactive metabolite - is associated with adverse out-comes, including higher 10-year CVD risk, higher prevalence of AO, increased glucose and HbA1c levels, increased lipid levels (TC, TG and LDL), higher SBP and lower HDL levels. These findings suggest that the cardiovascular risks of acrylamide are primarily mediated by its conversion to glycidamide.
Further longitudinal studies are needed to clarify the long-term effects of acrylamide on the cardiovascular system. The findings of this review, which highlight the potential adverse effects of acrylamide exposure, are of high public health relevance, as they provide evidence of associations with key cardiovascular risk factors and disease outcomes. While regulatory measures, such as those implemented in the EU and the US, represent significant progress, additional efforts are needed to increase public awareness of how home cooking practices contribute to acrylamide exposure. Educating consumers about strategies to minimize acrylamide formation, coupled with continued innovation in food industry practices, will be critical in addressing the broader health risks associated with this substance”.
- What measures were implemented to mitigate the heterogeneity across the studies regarding demographic, geographical location, and methodologies for acrylamide measurement in the synthesis of the results?
Author response: Thank you very much for helping us to make a better interpretation of the results. Heterogeneity was present due to substantial differences in study design, exposure measurement and outcomes analyzed, among others. This limited the possibility of making a formal meta-analysis. However, in order to reduce it, results were organized according to the main outcomes of interest, separating analyses for smokers and non-smokers when this information was available. Studies were also analyzed according to methods of measuring acrylamide exposure: hemoglobin adducts (HbAA, HbGA), urinary metabolites (AAMA, GAMA), and dietary estimates, highlighting differences inherent in each approach.
- Considering the inverse relationship shown between acrylamide exposure and some cardiovascular risk factors, can there be potential confounding variables that were insufficiently controlled in the studies?
Author response: Thank you again. Residual confounding could always be present in the included studies. Some important factors to be considered are diet and lifestyles (people with a diet high in acrylamide may have dietary habits associated with lower levels of other nutrients beneficial to cardiovascular health, such as antioxidants), and the interaction between acrylamide and other compounds (acrylamide-rich foods also contain other bioactive compounds that could positively or negatively influence cardiovascular risk factors).
Exposure to tobacco smoke is also an important source of acrylamide but an independent cardiovascular risk factor. Although models are often adjusted for smoking, passive exposure or differences in smoking intensity may not be adequately controlled. In addition, some genetic variants affecting the metabolism of acrylamide to glycidamide (mediated by CYP2E1) may modify both toxicity and association with cardiovascular risk factors (these differences are rarely assessed). No less important, socioeconomic factors do not fully capture the impact of socioeconomic status on access to healthy foods, lifestyles and medical care.
To emphasize the reviewer's comment, we have modified the discussion as previously shown.
- The significance of glycidamide in affecting cardiovascular risks is emphasized; what are the metabolic pathway differences between acrylamide and glycidamide that may elucidate their contrasting relationships with cardiovascular health outcomes?
Author response: Acrylamide is metabolized in the liver primarily through the CYP2E1 enzyme, which converts acrylamide to glycidamide. Glycidamide is much more electrophilic, which allows it to bind to proteins and DNA, causing greater genotoxic and mutagenic damage compared to acrylamide.
Acrylamide is mainly related to neurotoxicity and may have limited direct cardiovascular health effects due to its lower chemical reactivity. On the other hand, glycidamide has shown to induce oxidative stress, lipid and protein binding, which contributes to dyslipidemia, hypertension and vascular damage. In our review, HbGA was consistently associated with adverse effects such as increased glucose, triglycerides, blood pressure and abdominal obesity. The proportion of acrylamide converted to glycidamide varies according to CYP2E1 activity, influenced by genetic factors, age, alcohol consumption, diet and environmental exposure. This variability may explain why adverse effects are more evident in glycidamide.
To make that clearer, we have modified the following sentence in the introduction section:
“After exposure, acrylamide reaches the systemic blood circulation and is metabo-lized in the liver via the mercapturic acid pathway and partially metabolized by cyto-chrome P450 2E1 (CYP2E1) to the more reactive epoxide glycidamide [6]. Glycidamide causes greater genotoxic and mutagenic damage than acrylamide [7]. The proportion of acrylamide converted to glycidamide varies according to activity, which is influ-enced by genetic factors, age, alcohol consumption, diet and environmental exposure [6]. Both acrylamide and glycidamide are electrophilic and bind covalently to proteins such as haemoglobin (Hb) and DNA [8]. Hb adducts of acrylamide (HbAA) and of glycidamide (HbGA) provide a good estimate of long-term exposure. In contrast, the use of urinary mercapturic acid metabolites as biomarkers provides insight into short-term exposure to acrylamide (i.e., N-acetyl-S-(2-carbamoylethyl)-l-cysteine (AAMA) and N-acetyl-S-(2-carbamoyl-2-hydroxyethyl)-l-cysteine (GAMA)) [9]. The conversion of acrylamide to glycidamide is represented by the ratio HbGA/HbAA [10].”
- In light of the varied outcomes reported in the literature, how do the results of this review correspond with or diverge from current meta-analyses regarding acrylamide exposure and cardiovascular effects?
Author response: Most of the meta-analyses conducted to date have focused on the relationship between acrylamide gestational exposure and offspring growth, showing that maternal exposure is associated with a high risk of future overweight/obesity in these children. However, to our knowledge, there are no meta-analyses evaluating the cardiovascular effects of acrylamide.
- What are the consequences of the substantial dependence on data from the NHANES database, and how could this affect the generalizability of the findings to populations beyond the United States?
Author response: The reviewer mentioned an important point. Using data from NHANES is a strength for its data quality, but limits generalizability worldwide due to differences in diet, exposure and socioeconomic context. Results based on NHANES may overestimate or underestimate the risks associated with acrylamide in other countries due to differences in diet, environmental exposure and genetics. The lack of international comparative data makes it difficult to assess whether the observed findings are specific to the US or apply globally.
To emphasize the reviewer's comment, we have modified the discussion as previously shown.
- The review recommends conducting stratified analyses based on smoking status; what is the interaction between smoking and acrylamide metabolism, and what are the consequences for cardiovascular risk?
Author response: Smoking is a major source of acrylamide. Smokers have significantly higher levels of acrylamide biomarkers. Tobacco also induces the activity of the CYP2E1 enzyme, so smokers not only have higher levels of acrylamide but also higher conversion to glycidamide. In studies that do not stratify by smoking, the observed effects of acrylamide may be dominated by smokers, masking true associations in non-smokers or confounding the results. Non-smokers, whose main source of acrylamide is dietary, may have different cardiovascular risk patterns than smokers due to their lower overall exposure and less activated metabolism of glycidamide. As you can see in table 1, some of the studies were controlled for smoking status. In none of them the interaction for smoking status was considered.
- Given that cross-sectional research constitute the predominant evidence base, what particular longitudinal data would be essential to validate these apparent associations?
Author response: We agree with the reviewer on this point. The longitudinal studies may use validated blood biomarkers (HbAA and HbGA), which better reflect long-term exposure, at multiple time points during follow-up to capture changes in exposure and reduce misclassification errors; and record dietary patterns using validated questionnaires and repeat the assessment at regular intervals to reflect changes in diet. In the meantime, mitigation recommendations on industrial products should be maintained.
Summary of the changes made to the manuscript
- Revised one paragraph in the introduction section.
- Adjusted the order of measures of effect (OR, HR, and RR).
- Updated the title of Table 1.
- Replaced "Cerebrovascular disease (CeVD)" with "stroke".
- Corrected "Blood" in Table 1.
- Added a paragraph to the discussion section explaining the contradictory results.
- Included Table 8 in the discussion section to summarize the main conclusions of the review.
- Revised the limitations section.
- Updated the conclusions section.